# Overview of New Treatments with Immunotherapy for Breast Cancer and a Proposal of a Combination Therapy

**DOI:** 10.3390/molecules25235686

**Published:** 2020-12-02

**Authors:** Miguel Angel Galván Morales, Raúl Barrera Rodríguez, Julio Raúl Santiago Cruz, Luis M. Teran

**Affiliations:** Department of Immunogenetics and Allergy, Instituto Nacional de Enfermedades Respiratorias Ismael Cosío Villegas, Calzada de Tlalpan 4502, Sección XVI, Tlalpan, Ciudad de México 14080, Mexico; ygalvan2000@yahoo.com.mx (M.A.G.M.); barrerarr@hotmail.com (R.B.R.); juliosantiagocruz@yahoo.com (J.R.S.C.)

**Keywords:** breast cancer, immunotherapy, PD-1, CTLA-4, signal transduction inhibitors, PARP-1, autologous cells, and cytokines Th1

## Abstract

According to data from the U.S. National Cancer Institute, cancer is one of the leading causes of death worldwide with approximately 14 million new cases and 8.2 million cancer-related deaths in 2018. More than 60% of the new annual cases in the world occur in Africa, Asia, Central America, and South America, with 70% of cancer deaths in these regions. Breast cancer is the most common cancer in women, with 266,120 new cases in American women and an estimated 40,920 deaths for 2018. Approximately one in six women diagnosed with breast cancer will die in the coming years. Recently, novel therapeutic strategies have been implemented in the fight against breast cancer, including molecules able to block signaling pathways, an inhibitor of poly [ADP-ribose] polymerase (PARP), growth receptor blocker antibodies, or those that reactivate the immune system by inhibiting the activities of inhibitory receptors like cytotoxic T-lymphocyte antigen 4 (CTLA-4) and programmed death protein 1 (PD-1). However, novel targets include reactivating the Th1 immune response, changing tumor microenvironment, and co-activation of other components of the immune response such as natural killer cells and CD8+ T cells among others. In this article, we review advances in the treatment of breast cancer focused essentially on immunomodulatory drugs in targeted cancer therapy. Based on this knowledge, we formulate a proposal for the implementation of combined therapy using an extracorporeal immune response reactivation model and cytokines plus modulating antibodies for co-activation of the Th1- and natural killer cell (NK)-dependent immune response, either in situ or through autologous cell therapy. The implementation of “combination immunotherapy” is new hope in breast cancer treatment. Therefore, we consider the coordinated activation of each cell of the immune response that would probably produce better outcomes. Although more research is required, the results recently achieved by combination therapy suggest that for most, if not all, cancer patients, this tailored therapy may become a realistic approach in the near future.

## 1. Introduction

Cancer is one of the leading causes of death worldwide, and according to the status report of the global burden of cancer worldwide provided by GLOBOCAN, it will probably rise to 27.5 million new cancer cases and 16.3 million cancer deaths in 2040. Breast cancer (BC) is the second most commonly diagnosed cancer (11.6% of the total cases) and the leading cause of cancer death (18.4% of total deaths) [1]. This is the most common cancer in women, with approximately 2,088,849 new cases expected annually worldwide, accounting for nearly 1 in 4 cases of cancer among women [2]. The incidence of BC has been rising most rapidly in Africa, Asia, Central America, and South America, with 70% of cancer deaths taking place in these regions. Indeed, BC mortality rates are higher in many developing countries [3].

Breast cancer can be divided into four molecular subtypes: (1) Luminal A: Estrogen receptor (ER+), progesterone receptor (PR+), human epidermal growth factor receptor 2 (HER-2+) (and Ki67 < 14%]; (2) Luminal B (ER+, PR+, HER2+, and Ki67 > 14% or ER+, PR+, and HER2+); (3) HER-2 positive (ER-, PR-, and HER-2+), which represents 25% to 30% of breast cancers; and (4) the Triple-Negative Breast Cancer (TNBC: ER-, PR-, and HER-2-), accounting for 12% to 24% of all breast cancers. In addition, it is identified by the TNM system, the stage is based on the size and location of the primary tumor, the spread of cancer to nearby lymph nodes or other parts of the body, tumor grade, and whether biomarkers are present [4,5].

The greatest impact on BC treatment arose from the HER-2 overexpression in the cancer cells of some tumors. HER-2 is a member of the human epidermal growth factor receptor family (HER/EGFR/ERBB). The HER-2 oncogene is located on chromosome 17q12. HER-2, also known as c-erb2 or HER-2/neu, is comprised of four members: erb-B1, erb-B2, erb-B3, and erb-B4. HER-2 is a membrane tyrosine kinase receptor and, when activated, affects cell proliferation and survival. HER-2 amplification is the primary pathway of HER-2 receptor overexpression and has been shown to play an important role in the development and progression of certain aggressive types of BC, such as lobular cancer in situ (LCIS) or invasive ductal carcinoma (IDC) to a lesser degree. In recent years, this protein has become an important biomarker of cancer, and approximately 30% of breast cancer patients respond to antibody therapy [6,7]

The use of in patients receiving neoadjuvant chemotherapy (NAC) vs. adjuvant chemotherapy (AC) in both advanced and early-stage breast cancer (ESBC) has made significant progress with several landmark studies identifying clear survival benefits for newer therapies after radiotherapy, lumpectomy, or mastectomy. However, the use of anthracyclines and taxanes does not improve the prognostic disease, so therapy remains a major challenge. Increased improvements in the use of specific biomarkers (i.e., targeted RNAseq assay to measure both phenotype and genotype of HER-2+, ER+, PR+ related transcription, whether it is mutated or not) have resulted in significant results [8]. Moreover, multiple radiotherapy strategies are used to treat women at different tumor stages. For the majority of ESBC patients who are qualified for organ preservation, preoperative radiotherapy is a widely adopted standard intervention, whereas postmastectomy and radiotherapy are suitable for patients with advanced breast cancer. Nevertheless, not all patients undergoing radiotherapy benefit from it. Then and for both stages of BC, adjuvant and non-adjuvant endocrine therapy and/or ovarian ablation (releasing hormone antagonists GnRH, LH-RH, and others) are used to prevent the growth, spread, and recurrence of hormone-responsive cells [9].

For example, the selective estrogen receptor modulators (SERMs) like tamoxifen (Nolvadex^®^) and toremifene (Fareston^®^) (Figure 1) and others as fulvestrant (Faslodex^®^), palbociclib (Ibrance^®^), and mifepristone (Mifeprex^®^) are competitive progesterone receptor antagonist [10,11].

In BC, the causes of deaths are influenced by the occurrence of the disease, the availability of screening programs, suboptimal access to treatment, more aggressive biological subtypes, and younger age at diagnosis. It is estimated that one in six women diagnosed with BC will die. Cancer therapy is aimed at the total cure of the patient, so the best option is always chosen. In recent years, several major medical advances have increased the survival of cancer patients due to the valuable contributions made in the area of immunomodulators and targeted therapies. The first are molecules that act on the pathways that regulate the immune system’s activity, whereas the second are drugs or other substances that block or inhibit specific molecules involved in cancer growth, progression, and spread, using antibodies to block receptors and signaling pathways [12]. So far, several promising clinical strategies are being evaluated, including therapeutic monoclonal antibodies, immune checkpoint inhibitors, pathway inhibitors, and cell-based immunotherapy. Although there have been some good results with these strategies in the fight against BC, there is still a long way to go in terms of optimizing the effective treatment.

These new therapies are also being tested for in the aggressive TNBC since it does not respond to treatment with monoclonal antibody (MoAb) or anti-HER-2. This subtype of BC is defined by the absence of ER, PR, and HER-2 expression [13]. TNBC typically impacts younger women and is associated with increased relapse rates, more frequent metastasis, a worse clinical outcome with shorter overall survival (OS), and limited options of treatment [14]. The development of targeted therapies for TNBC is difficult because of its molecular heterogeneity. However, some preclinical studies of TNBC have identified some potential targets such as Src kinase, tyrosine-protein kinase Met (MET), and poly [ADP-ribose] polymerase 1 and 2 (PARP-1/2) [15], and more recently MYCN is a well-known oncogene identified from a sequence named v-gag-myc or v-myc for myelocytosis [16].

Thus, the development of new approaches for previous therapies and the success of new ones against breast cancer are expected to gradually reduce cancer mortality rates for the next decades. Here, we aim to review recent experience acquired, immunotherapy, and targeted therapy in BC and discuss how a combined therapy could be harnessed to (re)activate the immune response against cancer.

## 2. The Immune System and Cancer

The immune system has three main functions: to distinguish between the self and the non-self (foreign), to contain external invading agents (pathogens, molecules, etc.), and to destroy abnormal cells such as cancer cells. It is known that many tumors escape immune detection due to several factors such as a) lack of recognition of the HLA antigens, b) altered co-receptors that prevent their recognition, and C) T cell anergy or NK cells that allow or promote the development of cancer through “immunoediting”. As a result, many of the new anticancer strategies aim to redirect immune protection against these abnormal cells [17].

Encountering foreign antigens initiates the recognition of tumorous cells. This response is mainly by dendritic cells, natural killer cells (NK), macrophages, and lymphocytes B and T. NK cells are the most related to an antitumor response of innate immunity, as they eradicate tumors. Adaptive immunity is the main antitumor response that is aimed at specific receptors. Antigen-presenting cells are present throughout the body capture, process, and present tumor cell peptides to T cells. Here, the Th1 response mediated by interleukin (IL)-2, tumor necrosis factor (TNF)-α, IFN-γ, and cytotoxic T lymphocytes (CTL) will be activated. In addition, tumor cells that express the peptide will be recognized by NK cells, and then both cytotoxic T cells and NK cells will destroy the tumor that is found in the body; this response is specific since T lymphocyte can only recognize a single antigen, not others [18]. However, this does not always happen efficiently because most tumors do not efficiently express antigens, and also, these cells can shut down the reaction of T cells using PDL1 and CTLA-4 ligands. Some tumors can also grow without generating any signs of danger, resulting in permissibility and, in other cases, tolerance. Other control problems are immunodeficiency [19].

This highlights the fact that adoptive immunotherapy based on antibodies has not given the successful results expected due, in part, to the lack of accurate biomarkers in cancer. However, inhibitors of several signaling pathways or receptor-blocking antibodies, such as those directed against cytotoxic T-lymphocyte antigen 4 (CTLA-4) or programmed death protein 1 (PD-1), are being used to activate the antitumor response of T cells, with good results. Nowadays, immunotherapy strategies like chimeric antigen receptor (CAR) T cell therapy, cancer vaccines, and engineered T cell-based therapies and signal transduction inhibitors have been gaining regulatory approval and are in clinical development. Therefore, for a more efficient antitumor response, the changing tumor microenvironment increasing the degree of activation of the antitumor immune system will probably be necessary, which could be achieved through the stimulation of CTLs and NK cells, stimulated with cytokines such as IL-2, IFN-γ, TNF-α, and IL-12, plus the presentation of extracorporeal antigen [10].

## 3. HER-2 Targeting Therapeutic Antibodies

The main objective of immunotherapy is to recognize and eradicate tumors by restoring the immune capacity of the host. Spontaneous remission occasionally observed in malignant melanoma and renal cell carcinoma, even when the disease is quite advanced, offers hope that this goal is achievable [8]. The humanized monoclonal antibody (MoAb) trastuzumab (Herceptin^®^) recognizes the HER-2 receptor. The binding of trastuzumab to HER-2 promotes the increase of the p27kip1 protein that, in turn, stops cell proliferation [20,21]. Several clinical trials show that treatment with trastuzumab is more effective if combined with surgery and chemotherapy [22]. Trastuzumab added to polychemotherapy results in a 40% proportional and nearly 10% absolute overall survival (OS) advantage. A phase III clinical trial of aromatase inhibition (Anastrozole) with or without trastuzumab (TAnDEM study) among a subset of postmenopausal women with double, ER/PR positive and HER-2 positive cancers, found that the progression-free survival (PFS) was approximately 2 months on anti-estrogen alone and was doubled by the addition of trastuzumab [23].

Although adjuvant trastuzumab for early-stage I-III HER-2 positive BC has led to a noteworthy improvement in cancer outcomes, it carries a risk of cardiotoxicity. Therefore, trastuzumab is discontinued early in many patients for asymptomatic changes in left ventricular systolic function [24]. Furthermore, approximately 70% of patients develop trastuzumab resistance within a year [25], with no recognized biomarker to predict individuals who will become resistant. These adverse effects drove the necessity for the development of the new anti-HER-2 agents; lapatinib, pertuzumab, and trastuzumab emtansine (T-DM1) are some of them. Each of these drugs has demonstrated effectiveness in a different setting, especially with longer survival for HER-2-positive metastatic breast cancer (MBC). Thus, trastuzumab is included with chemotherapy or antiestrogens in the first-line therapy and, commonly, is re-incorporated later with other agents: ado-trastuzumab emtansine (KADCYLA) for the adjuvant treatment of patients with HER-2 positive early breast cancer (EBC) who have the residual invasive disease after neoadjuvant taxane and trastuzumab-based treatment (KATHERINE trial) [26] or lapatinib-containing regimens [27], in patients with HER-2 positive MBC [28]. Besides, trastuzumab deruxtecan has demonstrated very remarkable activity in patients with metastatic HER-2-positive BC, with an objective response rate (ORR) of almost 60% in patients that were previously refractory to T-DM1 (Table 1) [29].

Another human MoAb used in BC treatment is pertuzumab (Perjeta^®^) that inhibits the dimerization of HER-2 and HER-3 receptors; it is used in combination with trastuzumab as neoadjuvant therapy for BC at an early stage and, recently, also as adjuvant therapy to treat tumors in advanced stages (Figure 1 and Table 1) [30]. Results of a single large neoadjuvant trial NeoSPHERE demonstrated that pertuzumab added to trastuzumab plus docetaxel significantly increased pathologic complete response (pCR) in HER-2-positive EBC and locally advanced BC (LABC) [31]. The benefit of pertuzumab added to trastuzumab with an aromatase inhibitor (called THP) was demonstrated in the PERTAIN study, in which pertuzumab was effective for the treatment of HER-2 positive MBC/LABC, adding 3 months of PFS (hazard ratio (HR) 0.65; 95% CI, 0.48 to 0.89; *p* = 0.0070) to that of trastuzumab plus antiestrogens alone [32]. Therefore, based on its large survival benefit, THP is nowadays the gold standard therapeutic approach for HER-2 positive BC in the neoadjuvant setting and in the first-line treatment of metastatic disease. Trastuzumab and lapatinib (a dual EGFR/HER-2 tyrosine kinase inhibitor small molecule) can also be used in refractory patients with advanced disease [33].

Tumoral receptors targeted for immune therapy. From right to left, HER-2, and HER-3 receptors have a strong tendency to dimerize in cancer, *Pertuzumab* (green) (Figure 1). Antibodies colored in blue are *Nivolumab* (anti-PD-1) and *Ipilimumab* (anti-CTLA-4). *Bevacizumab* colored in dark green. HER-2 receptor is blocked by *Trastuzumab* (brown). *Cetuximab* (red) is an antibody directed to EGFR that inhibits signal transduction for cell growth. Other similar MoAbs are *Rituximab, Bevacizumab,* and *Ranibizumab*. Erlotinib (purple) is a drug that prevents the dimerization of EGFR receptors from heterodimer to homodimer. Tamoxifen and Fulvestrant (black) can inactivate estrogen action by their competitive antagonism. Mifepristone (grey) is an antagonist of the progesterone receptor, to be used in PR-dependent breast cancers. Lapatinib (red) is an inhibitor of tyrosine kinase from EGFR (ErbB1) and HER-2 (ErbB2) receptors. Other inhibitors such as *Saracatinib*, *Bosutinib*, or *Dasatinib* have also been used to avoid signaling on mitogen-activated protein kinase pathway (MAPK) and phosphatidylinositol-3-kinase (PI3K) pathway through Src or dimerization of Bcr-Abl tyrosine kinase. *Olaparib* is a molecule that inhibits PARP1. *Cobimetinib* is a reversible inhibitor of MEK that blocks MAPKs (yellow with purple) and *Vemurafenib* (red-yellow) inhibitors used for breast cancer metastatic.

Therapeutic agents such as bevacizumab (Avastin^®^), ranibizumab (Lucentis^®^), and aflibercept (Zaltrap^®^) (directed against vascular endothelial growth factor [VEGF]) are the first-line therapy for various retinal diseases, including neovascular age-related macular degeneration (nAMD) and diabetic macular edema (DME) [34], but their use in BC treatment in combination with chemotherapy is increasing [35]. A randomized phase III TANIA trial demonstrated that continuing bevacizumab with weekly paclitaxel as second-line chemotherapy for locally recurrent/metastatic HER-2 negative BC after progression on first-line bevacizumab-containing therapy significantly improved PFS compared with chemotherapy alone (HR 0.75, 95%; CI 0.61-0.93), although this did not translate into a significant OS advantage [36]. In patients treated with an anthracycline- and taxane-based chemotherapy, the application of bevacizumab as neoadjuvant chemotherapy (GeparQuinto trial) increased the pCR in patients with germline mutations in BRCA1/2 (gBRCA1/2) TNBC. However, their results were not significantly predictive of improved disease survival (DFS) in BRCA1/2-mutation carriers [37].

Neoadjuvant drugs such as gemcitabine, fluorouracil, epirubicin, docetaxel, doxorubicin, and cyclophosphamide have also been tested in combination with bevacizumab (BEVERLY-1) [38]. Unfortunately, a recent meta-analysis study failed to demonstrate the benefit of neoadjuvant bevacizumab in HER-2-negative non-metastatic BC [39].

Otherwise, the combination of palbociclib and fulvestrant seems to be a successful therapy in MBC. Fulvestrant is an ER receptor antagonist, while palbociclib is a CDK4/6 inhibitor that blocks the cell cycle [40]. The results of the survival analysis in phase III PALOMA-3 trial for HR-positive, HER-2 negative advanced BC showed that treatment with palbociclib–fulvestrant resulted in longer OS than treatment with placebo–fulvestrant (34.9 months, 95% CI: 28.8 to 40.0 vs. 28.0 months, 95% CI: 23.6 to 34.6; HR: 0.81; 95% CI, 0.64 to 1.03; *p* = 0.09) [41].

Recent findings indicate that the treatment of patients with basal inflammatory type BC or TNBC with anti-EGFR antibodies improves their prognosis [42]. Cetuximab (Erbitux^®^) is a chimeric MoAb approved in 2004 by the FDA for the treatment of human metastatic colorectal cancer, squamous cell carcinoma of the head and neck, and advanced non-small cell lung cancer. This MoAb inhibits EGFR activation by blocking its ligand binding or hindering its interaction with other proteins (Figure 1) [43,44]. Noticeably, the blockade with cetuximab also contributes to the suppression of several signaling pathways, such as JAK/STAT, Raf/MEK/ERK, and PI3K/Akt, which seems to be sufficient to inhibit the aggressive behavior in the TNBC [45]. However, a phase II trial of Cetuximab and Ixabepilone combination reported that clinical effectiveness was similar to Ixabepilone monotherapy in the first-line treatment of locally advanced or metastatic TNBC [46]. Similarly, the results of two additional independent randomized phase II clinical trials, evaluating the effects of Cetuximab in combination with either cisplatin or carboplatin [47], were similar to the clinical trial with Cetuximab and Ixabepilone (Table 1) [48].

Panitumumab (Vectibix^®^) is a human MoAb used primarily in the treatment of metastatic colorectal cancer and acts in the same way as cetuximab. Panitumumab is in phase III trial and is being used in combination with gemcitabine and carboplatin for treatment in women with TNBC [49]. Results of phase II clinical trial where panitumumab was combined with FEC 100 (5-fluorouracil, epidoxorubicin, cyclophosphamide 5-fluorouracil, epidoxorubicin/cyclophosphamide) and docetaxel as neoadjuvant therapy, showed an overall clinical response rate of 80% with acceptable toxicity in patients with operable TNBC [50]. Whereas, in a multicenter neoadjuvant phase II pilot study, the combination of panitumumab and cetuximab with taxane-anthracycline-containing regimens when treating operable TNBC showed a greater efficiency when compared to each treatment on its own [51]. This combination of panitumumab and neoadjuvant chemotherapy also showed improvement in the pCR rate in patients with primary HER-2-negative inflammatory BC [52].

**Table 1 molecules-25-05686-t001:** List of monoclonal antibodies approved for targeted therapy in breast cancer.

	Approved Agents	Molecular Targets	Mechanism of Action	Ref.
	Trastuzumab (Herceptin^®^)	HER-2, HER-3	RAS/Raf/MAPK Inhibition	[20,21,22,23,24,25,26,27,28,29,30,31,32,33]
	Pertuzumab (Perjeta^®^)	EGFR y HER-4		[26,32,53,54]
Monoclonal antibody	Bevacizumab (Avastin^®^)	VEGF	VEGFR PI3K, PLCγ prevents neovascularization	[34,35,36,37,38,39]
				[34,35]
				[55]
	Ranibizumab (Lucentis^®^)			
	Aflibercept (Zaltrap^®^)			
	Cetuximab (Erbitux^®^)	EGFR	PI3K, RAS, STAT signaling inhibition	[43,44,45,46,47,48,50,56]
	Panitumumab (Vectibix^®^)			[49,50,51,52]
Antibody-drug conjugate	T-DM1 (Trastuzumab emtansine)	HER-2	Cytotoxic agent vinca alkaloid	[26,27,57]
	Vinorelbine (Navelbine^®^) Trastuzumab (Herceptin^®^)		RAS/Raf/MAPK	[58]
	Sacituzumab govitecan-hziy		antibody-drug targeting trophoblast cell surface antigen 2	[59]

Zalutumumab is an IgG1 completely human anti-EGFR monoclonal antibody with the capacity to induce antibody-dependent cellular cytotoxicity activity (ADCC) [60]. In a randomized phase II trial, the combination of cetuximab plus cisplatin doubled the ORR and appeared to prolong PFS and OS compared to those treated with only cisplatin in patients with metastatic TNBC [56]. Besides, in a randomized phase II trial, the combination cetuximab plus carboplatin in metastatic TNBC produced responses in fewer than 20% of patients [48].

First-line treatment for BC. In personalized medicine, this treatment could be very effective since the administration of doxorubicin or irinotecan, both encapsulated in liposomes (green balls) to help their delivery into the site of action and diminish their secondary effects on other tissues, helps to avoid the worsening of the patient’s health (Figure 2). In the larger cell, the delivery of doxorubicin (or irinotecan) can be observed. In addition to antibodies against EGFR, such as Cetuximab, a PARP-1 inhibitor, such as Olaparib, could be added to avoid cells repairing the DNA damage. The activity of the PARP-1 inhibitor potentiates chemotherapy.

## 4. Signal Transduction Inhibitors

There are other inhibitors with the potential to be adopted in BC therapy: like those targeting the Src proto-oncogene family. Overall, in BC tumors with acquired drug resistance, the presence of large amounts of this activated Src is linked with a poorer outcome for the patients [61]. These drugs inhibit the signal transduction pathway that promotes critical processes for the development and progression of cancer such as proliferation, adhesion, invasion, migration, and tumorigenesis of cancer cells. The most important example, dasatinib (Sprycel^®^ or BMS-354825), a multi-target inhibitor approved for first-line use in patients with chronic myeloid leukemia, was developed in 2006 and was authorized for use by the FDA in 2013 [62]. This drug inhibits several tyrosine kinases, including Bcr-Abl, Src, c-Kit, and the Eph receptor, but does not affect EGFR [63]. Preclinical testing of SKLB646 (a potent inhibitor of Src, VEGFR2, B-Raf, and C-Raf) showed that it can be a promising lead compound for the treatment of TNBC. Additionally, by experimental models, it has been observed that the blocking of Src activity can reverse tamoxifen resistance [64].

However, despite several promising studies, a large number of clinical trials data using Src inhibitors in unselected MBC patients have been disappointing. A single-arm phase II clinical trial evaluated the combination of paclitaxel and dasatinib in patients with HER-2-negative MBC. Although the combination treatment did demonstrate activity in some patients, the study was halted early due to slow accrual [65].

In the recent phase II trial (GEICAM/2010-04) using dasatinib with trastuzumab and paclitaxel in first-line HER-2 positive MBC, the combination with trastuzumab-chemotherapy was more promising and reached an ORR of almost 80% [66]. Saracatinib (AZD-0530) is an experimental inhibitor of Src that acts as a dual inhibitor of kinases: Src and Bcr-Abl. However, despite several studies demonstrating its ineffectiveness against cancer, recently published articles still promote its use in cancer [67]. In a phase I trial, the application of saracatinib, the Scr-inhibitor, monotherapy in patients with double ER/PR-negative MBC does not appear to have significant activity [68]. Lapatinib (Tykerb^®^) is a tyrosine kinase inhibitor of HER-1/EGFR/ERBB1 and HER-2/ERBB2 and was authorized to be used in combination with capecitabine (Xeloda^®^). The usage of Lapatinib has given excellent results in the treatment of MBC and mainly in ER/EGFR/HER-2 positive BC that do not respond to trastuzumab, anthracyclines, or taxanes [69].

Phosphatidylinositol-3-kinase (PI3K)/AKT/mammalian target of rapamycin (mTOR) signaling pathway is one of the hallmarks in HR-positive BC. Activation of the PI3K/AKT/mTOR pathway not only is a critical step in oncogenesis but is also involved in resistance to anticancer therapies in ER/HER-2-positive BC. Mutations in PIK3CA (class IA) represents the most common genetic events in ER-positive BC, occurring at a frequency of 30% to 50%. Less commonly observed are mutations in PTEN (2% to 4%), AKT1 (2% to 3%), and PIK3R1 (regulatory subunit alpha; 1% to 2%). There is a higher frequency of PI3K/AKT/mTOR pathway mutations in BC, highlighting the potential impact for the implementation of targeted therapies. To date, there are multiple PI3K pathway inhibitors, which are subdivided into pan-PI3K inhibitors, isoform-specific inhibitors, and dual PI3K/mTOR inhibitors. Several are still being developed, but others are already being tested in clinical trials. For example, BELLE-2 is a double-blind, randomized, placebo-controlled phase III trial aimed to evaluate the use of buparlisib plus fulvestrant versus placebo and fulvestrant in HR-positive, HER-2-negative postmenopausal women. The results of this trial showed that the use of this PI3K inhibitor combined with endocrine therapy was associated with a significant improvement compared to fulvestrant alone. However, this combination was also found to have considerable toxicity, limiting its efficacy [70,71].

The BOLERO-2 study is a phase III clinical trial designed to evaluate the combination of everolimus and exemestane in the treatment of postmenopausal HR-positive advanced breast cancer. The results showed that the combination of everolimus (mTOR inhibitor) with exemestane increases the PFS (4 to 6 months) compared with exemestane plus placebo. However, an increased toxicity profile was also observed in the combination arm of exemestane and everolimus [72]. The phase III SANDPIPER (NCT02340221), a double-blind, placebo-controlled, randomized, phase III study evaluated taselisib (small molecule inhibitor targeting PI3K subtype PIK3CA) plus standard hormone therapy (fulvestrant) in postmenopausal women with ER-positive, HER-2-negative, PIK3CA-mutant locally advanced or MBC. Taselisib combined with fulvestrant significantly improved PFS (HR = 0.70) by 2 months longer than hormone therapy alone. Besides, the novel combination decreased the chance of cancer worsening by 30% [73]. Finally, results from the Phase III SOLAR-1 study showed that the combination alpelisib–fulvestrant prolonged the PFS in postmenopausal women with PIK3CA-mutated, HR-positive, HER-2-negative advanced BC who had received endocrine therapy previously [74].

Another possible combination to explore is the use of BRAF inhibitor dabrafenib (TAFINLAR^®^) plus Trametinib (Mekinist^®^), which is a reversible allosteric blocking of the MAPK pathway [75]. This cocktail is commonly used for melanoma treatment but has also been assessed in BRAF-mutant cancers and is expected to provide new therapeutic options in BRAF-mutated MBC [76].

Currently, other available agents include the ALK-inhibitors ceritinib and crizotinib and the EGFR inhibitors afatinib, erlotinib, and gefitinib. Four oral targeted therapies are used in the treatment of solid tumors associated with the B-Raf proto-oncogene (BRAF), cobimetinib, dabrafenib, trametinib, and vemurafenib; all are oral agents for treatment (Table 2 and Table 3) [77].

However, these and several other clinical trials have shown that the efficacy of these types of inhibitors as therapeutic agents is far from that expected due to the coexistence of various mutations present in tumor cells, or the toxicity associated with these treatments.

## 5. Checkpoint-Blocking Antibodies

It has been observed that in some tumors, cytotoxic T cells are repressed and cannot respond against the tumor through the interaction between T cell receptors and co-receptors [PD-1 and its corresponding ligands PD-L1/PD-L2 or B7.H1/B7.DC, and CD152/CTLA-4 and its ligands CD80 (B7-1) and CD86 (B7-2)] present in the cancer cell. The subtypes, HER-2+ breast cancers, and TNBCs are more likely than luminal breast cancers to harbor stromal infiltrating immune cells and are also more likely to express the programmed death ligand-1 (PD-L1). Clearly, the use of these checkpoint-blocking antibodies has shown great therapeutic potential in several clinical trials, which is the reason they have been proposed for the first-line treatment of BC [104]. Recently, in phase I/II open-label trial of with nivolumab (anti-PD-1mAb) plus ipilimumab (anti-CTLA-4-mAb) in TNBC (NCT01928394), and in a single-arm phase II trial of durvalumab plus tremelimumab (anti-CTLA-4-mAb) for HER-2 negative MBC (NCT02536794), the results suggest improved clinical responses by checkpoint blockade therapy due to their high immunogenicity observed in these cancer types. Atezolizumab (Tecentriq^®^), a humanized monoclonal antibody of the isotype IgG1 against PD-1, and pembrolizumab (Keytruda^®^), which act by blocking PD-1, are also available but they should not be used in immunosuppressed or treated patients with a corticosteroid (Figure 3a,b) [95].

Restoring the immune capacity. Binding of PD-1 with its ligands B7.H1 and B7.DC delivers inhibitory signals to T lymphocytes (Figure 3a), this leads to anergy and dysfunction of cells. Both, activation mechanisms of T cells by antigen-presenting cells (APC) and blocking by MoAb targeted to PD-1/2 (red) and/or MoAb directed to CTLA-4 (blue), will restore the ability of T lymphocytes for destroying cancerous cells, especially if accompanied by the action of IFN-gamma and other cytokines. b) The PD-L1 interaction of cancerous cells favors the conversion of bystander T cells and the decrease of cytokine secretion. The MoAb anti-PD-L1 (green) binding to tumor cells avoid the inactivation of T cells and further can function as a marker for the attack to the membrane by the MAC, CTL-mediated death by perforin and granzyme, or NK CD16 cells with its Fc receptor that recognizes the antibodies bound to the cell. Nonetheless, there are no antibodies targeted to ligands of CTLA-4 (CD152 or CD137).

Polk et al. conducted a meta-research of the studies for breast cancer treatments until 2017; it highlights the results obtained that immune checkpoint inhibitors (durvalumab-tremelimumab and nivolumab-ipilimumab) with chemotherapy produce durable tumor remission, increase pCR, and induce long-standing antitumor immunity in a subgroup of heavily pretreated TNBC patients [101]. Other studies have begun the evaluation of the combination of durvalumab-tremelimumab in early-stage BC, as the treatment in MBC and TNBC has demonstrated good results [102,108]. Another anti-PD-L1 antibody being studied in a clinical trial against breast cancer is avelumab (Bavencio^®^). This human IgG1 anti-PD-L1 mAb also triggers ADCC against cancer cells and could be combined with chemotherapy or as a neoadjuvant setting prior to surgery [106].

While targeted therapy anti-PD-1/PD-L1 has made great strides in the treatment of BC, it simultaneously has been employing anti-CTLA-4 blocking antibodies as ipilimumab (Yervoy^®^). This inhibits CTLA-4 activation, which renders cytotoxic T lymphocytes capable of recognizing and destroying cancer cells. A single-arm, multicenter phase II clinical trial (NCT03789110) is evaluating the combination of nivolumab plus ipilimumab in mutated HER-2-negative MBC [96]. Together, nivolumab and ipilimumab are being tested, and their clinical results have been promising in the treatment of several cancers (Figure 1 and Figure 3a) (Table 2) [97].

## 6. Cytokines and NK Activation

In the early days of immunotherapy against cancer, some cytokines that regulate the Th1 response (IFN-γ, IL-2) were used together with nivolumab and ipilimumab. These cytokines can promote a better antitumor response after activating the T cells since the activated cells can recognize tumor antigens while these antibodies would release the brake imposed on the reactive T cells [107]. In experimental studies in vivo and in vitro, the use of these therapies has been favorable with the administration of IFN-α or IFN-γ and exogenous IL-2 (to activate the antitumor T cells) and the presence of regulatory T cells (Treg) or helper T cells 17 (Th17). Although the results of retrospective studies with patients who were treated with high doses of cytokines were not as encouraging, the combined use of cytokines and checkpoint-blocking antibodies may help boost the immune response against cancer cells and improved RFS and OS in patients with early BC [109]. Furthermore, these substances can be administered in combination with stage IV BC chemotherapy. Although side effects with the use of cytokines may be a limiting factor, careful management by an immunotherapy medical specialist can significantly reduce the risks [110,111].

NK cells are the main cellular effectors of the innate immune system, which mediate the lack of self-recognition, lysis of a marked target and are also a potent and early source of cytokines and chemokines, which does not require prior exposure to antigens. In healthy individuals, 90% of NK cells are mature and cytotoxic peripheral blood cells, characterized by expressing CD16bright and CD56dim, while the remaining 10% of NK cells are a subset of immature cells, which produce cytokines and express CD56bright CD16dim or CD16- and CD25+ [112]. The activity of NK is regulated by inhibition or stimulation of signals upon contact with tumor cells, where the prevalence of stimulating signals or the absence of inhibitors leads to triggering the lytic effect. That is, if healthy cells do not express high levels of HLA class I, and do not express activating ligands, tolerance will be induced in NK cells [113]. When tumor cells show a decrease in the expression of HLA class I molecules, there is an engagement of activating receptors such as NKG2D, DNAM-1, and the natural cytotoxicity receptors NKp30, NKp44, and NKp46, which trigger a robust activation of NK cells and NK cell-mediated killing of transformed cancer cells [114,115].

During the last decade, it was established that NK cells are activated by IL-12, IL-15, and IL-18, but the NK lineage is sufficiently exclusive since it is not easy to obtain functional NK cells. Colucci et al. proposed three stages in the development of NK cells in vitro. Under these conditions, NK cells show cytotoxicity and are producers of proinflammatory cytokines such as IFN-γ, TNF-α, IL-13, IL-10, and GM-CSF, when there is contact with tumor cells susceptible to be recognized. However, in most cancer patients, NK cell populations are depleted or almost eliminated or non-existent, resulting in low control of tumor growth [116]. It should be mentioned that the recognition of the tumor by NK cells can be carried out through an activation signal or the inhibition of the receptors on the surface of the cell, either measured by the MHC, by their KIR receptors, or they can be activated through an event independently of this recognition, and known as CIK (killer cells induced by cytokines).

Several reports have addressed the importance of NK cell effectors in treating certain tumors because of their ability to evoke ADCC. However, to obtain an effective antitumor immune response, the state of lymphocyte activation during treatment must be determined since the success of the treatment depends on it [117]. Thus, it has been reported that therapeutic anti-HER-2 MoAbs are capable to evocate an ADCC activity helpful in the BC treatment [118,119]. This occurs because when the MoAbs are binding to HER-2 positive tumors, there is an increase in the densities for FcγRIIIA sites (CD16A) [79]. Therefore, the treatment of HER-2-positive BC with trastuzumab and pertuzumab in the neoadjuvant setting and the first-line treatment for metastatic disease trigger ADCC activity and indirectly enhance the development of tumor-specific T cell immunity [53]. Besides, in patients that have progressed prior to trastuzumab, pertuzumab, and T-DM1, the treatment with lapatinib seems to increase the possibilities for NK cell-mediated ADCC antitumor responses [57]. Noteworthy is that despite tumors with gene expression signatures associated with the presence of cytotoxic lymphocyte infiltrates benefit from trastuzumab-based treatment, NK cell-related biomarkers of response to HER-2 therapeutic antibodies remain elusive [54,119].

NK cells also express classical checkpoint receptors, including PD-1, CTLA-4, TIM-3, TIGIT, and LAG-3 [120,121]. In particular, PD-1 expression has been found in CD56dim NK cells, and CD56neg NK cells, whereas CD56bright subset is consistently PD-1-negative. Interestingly, the PD-1-positive NK phenotype correlates with an impaired NK cell activity against PD-L1 expressing tumor cells, which can be partially restored by MoAb-mediated disruption of PD-1/PD-L1 interaction. Furthermore, NK cells have the potential to improve anti-PD-L1 MoAb therapeutic efficacy through ADCC and enhance the antitumor response against high PD-L1-expressing tumors, rendering this a beneficial strategy to overcome resistance. All these are important features because the use of anti-PD-1 or anti-PD-L1 MoAbs may, therefore, increase their activation state and cytolytic abilities, particularly against HLA-I-deficient tumor cells. Various clinical trials are underway investigating the effects of checkpoint inhibitors on NK cells [122,123]. One of them involves the use of anti-NKG2A (monalizumab) or anti-KIR (lirilumab) antibodies as a combo therapy with anti-PD-1 (nivolumab) in order to obtain a complete reconstitution of the antitumor NK cell cytotoxic response [98]. Therefore, we believe that the use of cytokines to activate autologous NK cells in vitro and then to be perfused back into the patient, so that they can respond against cancer cells, could be excellent immunotherapy (Figure 3a,b) [124].

## 7. Other Inhibitors Used in Chemotherapy

As mentioned above, many regulatory mechanisms participate in the development of TNCB. One of them is associated with germline mutations in BRCA1/2 (gBRCA), this alteration occupies more than 15% of TNBC, as well as it has been proposed that they could be sensitive to DNA-damaging agents such as cisplatin, carboplatin, and poly (ADP-ribose) polymerase inhibitors (PARPi). Consequently, in 2018, the FDA approved the use of olaparib (Lynparza) as neoadjuvant therapy for patients with gBRCA mutations, HER-2 MBC-negative status (Figure 2). Recently, a phase III OlympiAD trial in patients with MBC and BRCA1/2 mutation and HER-2-negative status showed a statistically significant increase in PFS but no OS benefit (Figure 1) [125].

Another mechanism associated with the development of TNBC is the nuclear localization of geminin (GMNN). Geminin is a coiled-coil protein that plays a critical role in preventing abnormal DNA replication by binding to and inhibiting the essential replication factor Cdc10-dependent transcript (Cdt1) [126]. A conformational change between GMNN–Cdt1 complexes is responsible for licensing or inhibition of DNA replication. GMNN interacts with different proteins, one of them is the c-Abl non-receptor protein tyrosine kinase. It is noteworthy that in solid tumors, activation of c-Abl kinase promotes invasion of BC cells and regulates responses to oxidative stress and DNA damage, cell proliferation, and survival [127]. The binding of c-Abl to GMNN reduces the stability of the GMNN–Cdt1 complex, which alters DNA stability, generates aneuploidy, and prevents apoptosis. Taken together, these studies suggest that the pathway involving the functional interaction between c-Abl and GMNN plays an important role in breast carcinogenesis. An analysis of >800 samples of TNBC showed overexpression of GMNN in ~50% of samples, all of them with overexpression of c-Abl located in the nuclear region. In the tumor samples with no GMNN expression, the protein c-Abl was only cytoplasmic. In this study, when the activity of c-Abl was inhibited by imatinib (Gleevec^®^) or nilotinib (Tasigna^®^), the phosphorylation of geminin in the Y150 residue was prevented, inactivating it and converting the overexpressed geminin into an inducer of apoptosis [128].

Preclinical studies identified poly [ADP-ribose] polymerase 1 and 2 (PARP-1/2) as potential therapeutic targets for TNBC. The effectiveness of PARPi, a group of pharmacological inhibitors of the enzyme PARP, relies on cancer cells undergoing oxidative stress, increasing DNA damage, and depletion of cellular ATP that leads to lysis and cell death (necrosis) [129]. Therefore, new strategies in cancer therapy seek to inhibit PARP activity because knock-out mice for PARP-1 and -2 showed profound deficiencies in the mechanisms of DNA repair, as well as greater sensitivity to ionizing radiation or alkylating agents [130]. Some examples of PARPi are iniparib (Sanofi^®^ BSI 201), talazoparib (BMN-673), olaparib (Lynparza^®^ AZD-2281 and TOPARP-A), veliparib (ABT-888), and niraparib (Tesaro^®^) (Figure 1) (Table 3) [131,132,133,134,135].

**Table 3 molecules-25-05686-t003:** Poly [ADP-ribose] polymerase (PARP)-1/2 inhibitors and other signal pathways inhibitors in breast cancer treatment.

	Approved Agents	Molecular Targets	Mechanism of Action	Ref.
Inhibitors PARP-1/2	Iniparib (Sanofi^®^ BSI 201)	Poly (ADP-ribose) polymerases (PARPs), especially PARP1, PARP2 selective inhibitor	Not repair their DNA breaks sites	[130,135,136]
	Talazoparib (BMN-673)			[131,137]
	Niraparib (Tesaro^®^)			[134,135]
	Olaparib (Lynparza^®^ AZD-2281 y TOPARP-A)			[125,132,138,139]
	Rucaparib (AG014699, PF-01367338)			[140]
	Veliparib (ABT-888)			[131,133,139,141]
	CEP-9722	PARP-1/2 inhibitor	Not repair their DNA	[142]
	NMS-P118			[143]
	AG014699			[144]
Tyrosine Kinase Inhibitors and other signal pathways	MK-2206	Serine/threonine kinase Akt	Allosteric Akt inhibitor (Cell cycle arrest)	[145]
	Dasatinib (Sprycel^®^)	Multiple tyrosine kinases (TK)	Bcr/Abl, Src, c-Kit and Eph receptor family	[61,62,63,64,65]
	Bosutinib	Src tyrosine kinase	ATP-competitive Bcr-Abl tyrosine-kinase inhibitor	[146]
	Saracatinib (AZD0530)	Src protein	Src inhibitor Bcr/Abl	[67,68]
	Imatinib (Gleevec^®^)	Geminina y c-Abl nuclear	Inhibitor Geminina y c-Abl nuclear	[128]
	Nilotinib (Tasigna^®^)	Abl tyrosine kinases	Inhibitor Bcr/Abl	[147]
	Lapatinib (Tykerb^®^)	TK receptor (HER-2)	Inhibits the tyrosine kinase activity	[22,27,28,33,57,69,79]
	Palbociclib (Ibrance^®^)	Cell cycle (CDK)	Inhibitor kinases (CDK4,6)	[11,40,41,148]
	Ribociclib (Kisqali^®^)			[149]
	Cobimetinib (Cotellic^®^)	MEK1 protein kinase	Inhibits the activity of ERK2 transcriptional	[150]

In 2014, Sanofi gave up the use of iniparib after phase III trial in TNBC results were reported on the “first Word pharma” portal. Based on phase II clinical trials in MBC and TNBC, the use of PARPi including iniparib is being recommended as the first-line treatment of these cancer types [136]. Olaparib was approved by the FDA in 2019 for the treatment of gBRCA1/2 mutated, HER-2-negative MBC with previous chemotherapy in the neoadjuvant, adjuvant, or metastatic settings. Results of the REVIVAL trial provided a significant benefit of olaparib monotherapy over standard therapy; the median PFS was 2.8 months longer and there was a risk of disease progression [138]. Moreover, the approval of talazoparib was based on the EMBRACA open-label phase III trial in which patients with MBC and a gBRCA1/2 mutation to receive talazoparib or standard therapy. Median PFS was significantly longer in the talazoparib group than in the standard therapy group (8.6 vs. 5.6 months; HR 0.54, 95% CI 0.41–0.71; *p* < 0.001). The ORR was also significantly higher in the talazoparib group (62.6% vs. 27.2%; OR 5.0, 95% CI 2.9–8.8; *p* < 0.001) [137]. Both PARPi are generally well tolerated. In OlympiAD, REVIVAL, and EMBRACA trials, only about 5% of patients stopped treatment because of adverse events. The most common side effects associated with both olaparib and talazoparib are anemia, nausea, and fatigue. Veliparib has been used along with chemotherapy based on liposomal doxorubicin (Caelyx^®^, Myocet^®^) [139], or nanoliposomal irinotecan (Onivyde^®^), and recommended as first-line treatment in TNBC (Figure 2 and Table 3) [141,151].

## 8. A Proposal of Combination Immunotherapy

Immunotherapy with antibodies does not always activate the immune system efficiently. This is because of the functional heterogeneity of immune response within different areas of the same tumor. Therefore, it has been necessary to find strategies to improve the immune response to BC. One interesting approach can be the co-activation of macrophages (MΦs) (retinoic acid), NK cells (IL-15, IL-22, and IL-23), cytotoxic T cells (CTLs) (IL-2, TNF-α, IFN-γ), and the acquisition and presentation of tumor antigens by dendritic cells [152]. Accordingly, the next step in cancer therapy could be depletion or maintenance of the Treg/Th17 cells balance [153].

So, an approach that we propose to boost the anticancer response of the immune system is to “tag” cancer cells with specific antibodies, and activate both the antigen presentation and lymphocytes with Th1 cytokines (IFN-α, TNF-α, IL-2, IFN-γ, and IL-12). To minimize the promoted attack, if it does not physiologically stop, IL-10, TGF-β, and autologous Treg would be applied to restore equilibrium (Figure 4) [154]. In this experimental proposal, autologous CD4+ T cells could be robustly polarized in vitro toward Th1 and Th17 subtypes, through cytokines and chemokines, to subsequently identify their activation with molecular adhesion profiles and surface markers, which will open the possibility of detecting effector function in vitro and then in vivo to treat cancer.

Th1 response is generally considered as the main source of tumor rejection, but there is also evidence that Th17-polarized cells mediate the destruction of advanced B16 melanoma. The Th1 therapeutic effect critically depends on the production of IFN-γ, while apparently depletion of IL-17A and IL-23 has little impact. Visibly, the appropriate in vitro polarization of the effector CD4+ T cells could be decisive for the elimination of the tumor. Design of preclinical and clinical trials should consider this therapy based on in vitro stimulation of the response and posterior transferring of these immune cells, capable of recognizing the tumor in vivo, back to the original tumor (Figure 5) [155].

The evolution of cancer therapies does not appear to improve the prognosis by that only a partial resolution of the problem is achieved. In the worst-case scenario even with several types of treatment, only a few more years of life are provided to the patient and, in some cases, they produce severe adverse reactions. The autoimmune response is a frequent complication that, besides, prevents the adequate identification of the tumor. This event raises several questions: Why do tumors escape immune surveillance? Is tolerance due to the early presence of cancer cells, evading, therefore, immune recognition of the self and nonself? Lack of response is due to molecules that fulfill their role abnormally? Does immunoediting happen in all cancers?

The processes carried out by the immune system to recognize and eliminate a tumor are explained by immunoediting that comprises three stages: elimination, equilibrium, and escape. If in the elimination phase the tumor is not controlled, it enters a stage of equilibrium, and the escape will be allowed. Furthermore, there is a possibility that the recognition of antigens does not occur, there are T cell anergy, or the activation of cytokines is not carried out.

Extracorporal activation of autologous cells to stimulate Th1 and NK responses (Figure 4). Several lineages of bystander immune cells are obtained from the cancer patient to be activated. T cells are challenged with chimeric antigens. Further, the experimental in vitro boost of tumoral cells with NK cells and T cells costimulated with interleukin accordingly to the lineage for the maturation and differentiation of immune cells. Later, theoretically stimulated cells are reincorporated to attack the tumor. (1) Tumor tissue is removed by a patient biopsy. (2) The patient’s tumor tissue is cultured in-vitro exposed to cytokines and antigen-presenting cell (APC). (3) After the activation and presentation, cell cytokines are collected from immune cells. (4) Everything is inoculated to the tumor.

The same tumor piece can be embedded into a non-degradable synthetic polymeric tube with hexagonal holes (Figure 5). This encapsulated tumoral piece is going to be embedded with Th1 cytokines and chemokines IL-2, IFN-alpha or -gamma, and TNF-alpha to activate the cell-mediated cytotoxicity, and the NK cells with IL-12, IL-15, and IL-18; at the last, the encapsulated piece reincorporated. Monitoring of the maturation of response and the new immune awakening should be screened with the attack to other sites with a free tumor. (1) Population NK, macrophages, dendritic, and CD8+ T cells on their local tissue residency recirculating in the peripheral blood. (2) Autologous cell activation in vitro, with invasive protein/antigen or directly on cancer cells. (3) Cancerous tissue and autologous cells, suffused with cytokines. (4) Tumor tissue is encapsulated in a hexagonal opening structure, embedded with cytokines, interleukins, and immunity cells. (5) Encapsulated tumor tissue is introduced at the site where the biopsy was removed or in the peritoneum where there is a good immune response.

With the acceptance of the heterogeneity observed in BC subtypes and the molecular mechanisms that contribute to the emergence of treatment resistance and metastatic disease, the implementation of more effective therapeutics is necessary to increase the rate of survival of BC patients. Currently, immunotherapy based on antibodies is being incorporated into almost all cancer treatments. Yet, it seems rational to combine MoAb therapies with other agents and regimens like a PARPi such as olaparib (Lynparza^®^) and liposomal chemotherapy. This combination therapy appears to be promising in the treatment against cancer (Figure 2) [138,156]. Therapeutic combinations could include: (i) the use of the chemotherapy encapsulated in liposomes (i.e., CAELYX^®^). The application of CAELYX^®^ plus spores of *Clostridium* (spores of *C. novyi-NT*) in the site of the tumor or metastasis will aid in situ release [157,158]. (ii) A least one, a PARPi or an Src inhibitor, and (iii) Additionally, the application of therapeutics antibodies directed to target molecules could be necessary that block the proliferation of cancer cells and promote cell cytotoxicity, together with specific inhibitors for overexpressed receptors TKi. Furthermore, the correct implementation of this combination therapy will be recommendable to verify the production/expression of TNF-α, IFN-γ, and IL-2. This triple approach for BC treatment could be a better option compared to first-line treatments with EGFR inhibitors alone or with standard chemotherapy, which have ineffective response rates (Figure 2).

Our proposed immunotherapy against tumors also includes the reactivation of the immune response through a new screening and activation, using activated T cells and external cytokines, which may be re-implanted in the patient to establish an effective and lasting recognition of the tumor. This proposal might be validated as a common treatment against several types of cancer since it is based on the reactivation of the immune system for the recognition of new variants and the activation of the immune antitumor response. Briefly, our first proposal would be extracting naïve T lymphocytes and tumor cells from the patient, and co-cultivate them in the presence of cytokines to activate cytotoxic T-cell response: i.e., IL-2, INF-α and γ, and TNF-α. Additionally, MΦ or dendritic cells should be obtained; they could be activated in a classical way (M1) to be lately exposed to cancer cell/tumor-associated antigens obtained of cancer cells arise from one’s own tissue to carry out the antigen processing. An additional possibility is that excised tumors of patients could be confronted with *ex vivo*–expanded autologous NK cells and exogenous cytokines IL-15, IL-18, IL-22, and IL-23 to reactivate the immune response and establish specific tumor recognition. As a result of all these interactions, T-cells and antigen-presenting cells already activated could be re-implanted into the patient to raise an efficient and sustained antitumor immune response (Figure 4). A second immunotherapy approach consists of tumor extirpation to infiltrate it later with Th1 cells and IL-15, IL-18, and IFN-α cytokines and subsequently reimplant the tumor-tissue into the patient, in a subcutaneous region closer to some lymph nodes intraperitoneal. It must be noted that this implant should be covered in a nanoparticulate network to prevent the escape of tumor cells (Figure 5). As a result, a reactivation of the local immune response mediated by T cells, antigen-presenting cells, and NK cells is expected. Through the active migration of immune response cells to the implanted tumor, the stimulated area might be monitored.

In the fight against cancer, immunotherapy success will only be achieved through a deep understanding of the T cells and NKs activation mechanisms, and delicate handling of cytokines, reaching adequate cytotoxic T cells reprogramming. All this additionally depends on the uptake and presentation of tumor-associated antigens by MΦ and dendritic cells, coupled with the expression of various co-stimulatory molecules and cytokines of the antitumor response. Once established, it can be kept under control by regulatory mechanisms, such as immune checkpoint molecules (PD-1 or CTLA-4), as well as other types of immunosuppressive cells such as Treg and TH17.

## 9. Conclusions

The current standard treatment for different types of BC is still chemotherapy since they are highly sensitive. However, tumor resistance and recurrence results in low rates of survival. Thus, the design of several MoAbs directed to tumor antigens has proven successful in the treatment of some type of BC. Besides, the immune checkpoint blockade is playing a preponderant role in the immunological awakening. Other therapeutic strategies to consider are: (1) the TKi tests have shown that, despite the preservation of tyrosine kinase domains, tyrphostins can be designed and synthesized that discriminate even between closely related tyrosine kinase proteins, such as EGFR, and its close relationship with HER-2 and thus can be used in breast cancer that expresses it; (2) the use of inhibitors for AKT, Scr, and other downstream signal molecules involving in the activation of genes promoting cell proliferation and differentiation; (3) The use of inhibitory molecules for several inhibitors mitogen-activated protein kinase pathway (MAPK/ERK) or PARP-1/2 causes multiple breaks in double strands and tumors with BRCA1, BRCA2 present in some tumors of BC.

With the recent advances in antitumor immunology and the development of new inhibitors of target molecules, the implementation of combination therapy against cancer seems to be the most promising option in years to come. This goal will only be reached with further research, increasing the number of clinical trials, and obtaining the experience that will allow the future achievement of effective therapy against one of the diseases that cause many deaths in the world [148,159,160].

This is an empty area that needs to be filled in soon. We propose the future clinical trials should consider a careful analysis of immune cell gene signatures and molecular single-cell spatially resolved data from clinical samples, which will provide valuable information over the determinants of response to therapy, and drive the best treatment to follow. In the near future, clinical trials with combination therapy could revolutionize the care of early-stage and metastatic BC, and ideally improve survival rates.

## Figures and Tables

**Figure 1 molecules-25-05686-f001:**
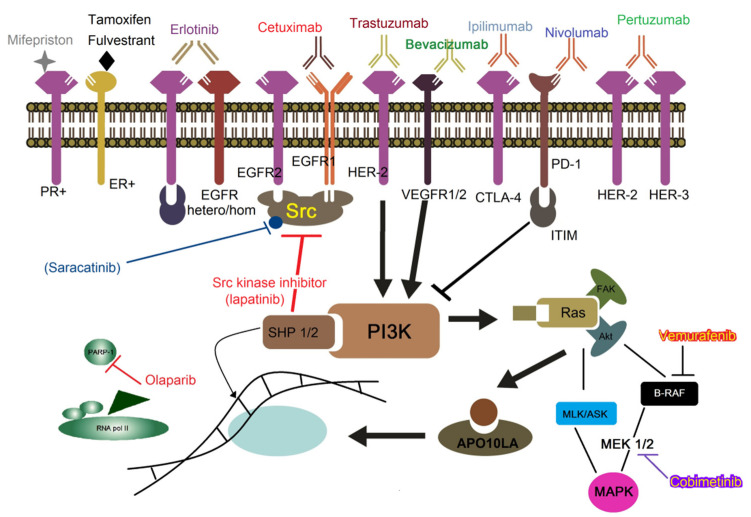
Therapeutic monoclonal antibodies and inhibitors approved for use in oncology.

**Figure 2 molecules-25-05686-f002:**
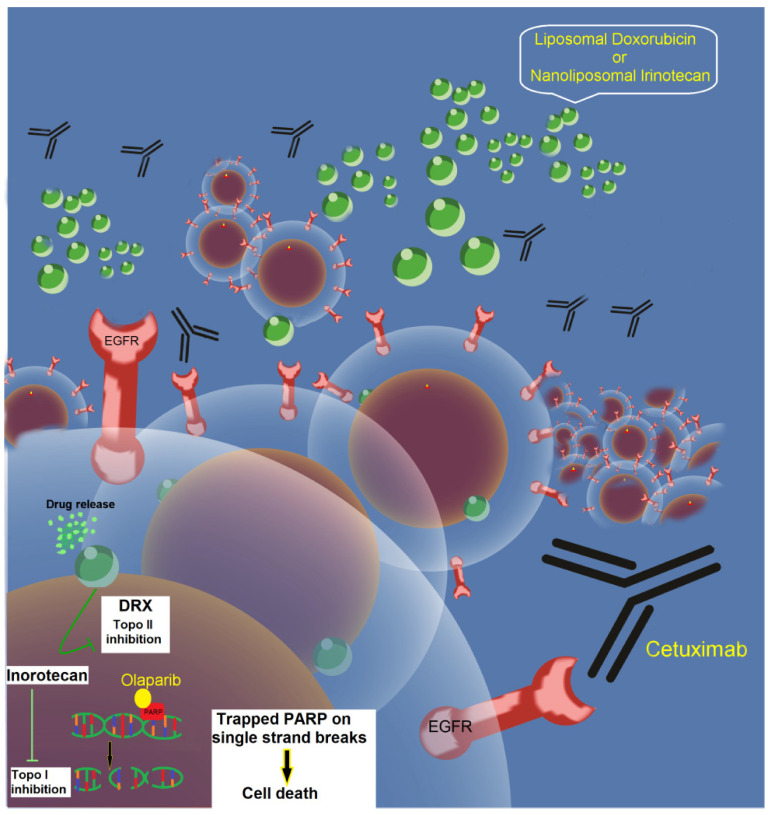
Combination therapy proposal.

**Figure 3 molecules-25-05686-f003:**
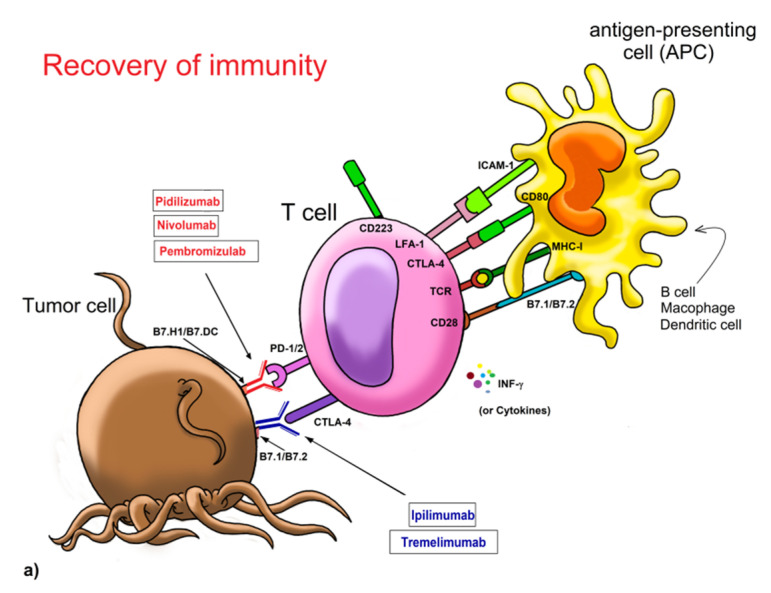
(**a**) Immunotherapy. (**b**) Lack of immunotherapy.

**Figure 4 molecules-25-05686-f004:**
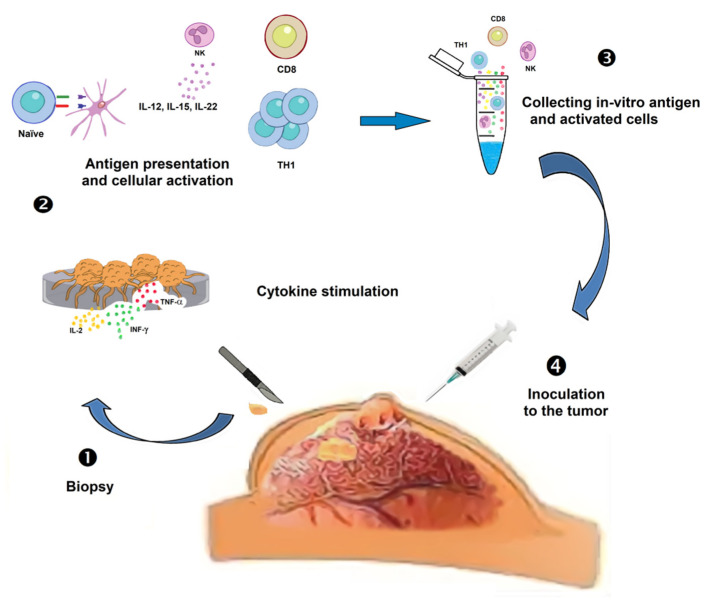
Response activation.

**Figure 5 molecules-25-05686-f005:**
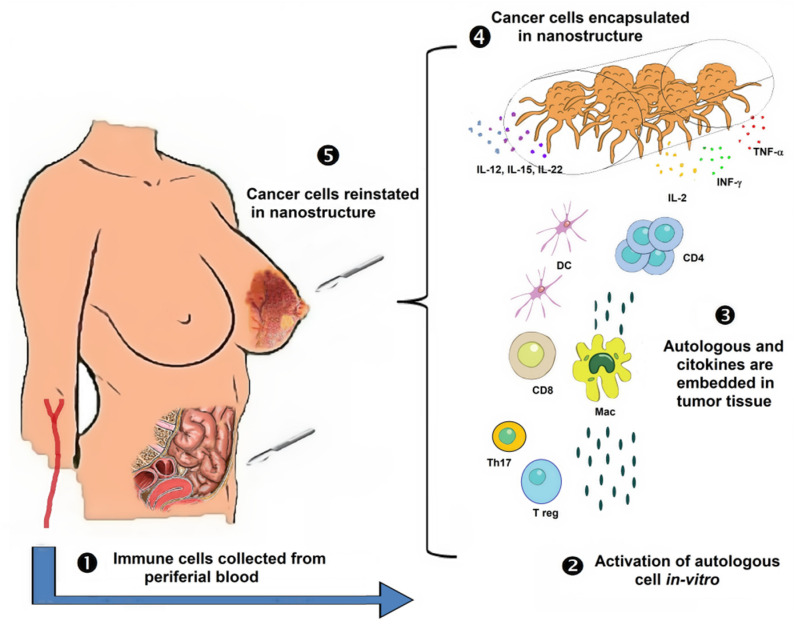
Activation autologous cell.

**Table 2 molecules-25-05686-t002:** Inhibitors and immunomodulators used in breast cancer therapy.

	Approved Agents	Molecular Targets	Mechanism of Action	Ref.
EGFR Inhibitors	Afatinib	Inhibits EGFR (ErbB1), HER2 (ErbB2), and HER4 (ErbB4) receptors	Irreversibly binds to members of the ErbB receptor family	[78]
	Lapatinib		Inhibit the intracellular phosphorylation of tyrosine kinase associated with both wild-type and mutation EGFR	[57,79]
	Erlotinib	Inhibitor of EGFR tyrosine kinase		[77,80]
	Gefitinib	Inhibitor of EGFR tyrosine kinase	Selectively binds to the EGFR-tyrosine kinase domain, preventing ATP from binding and blocking subsequent receptor autophosphorylation	[81]
	Dacomitinib	Inhibitor of the activity of the human EGFR family	Inhibition via irreversible binding at the ATP domain of the EGFR proteins (EGFR/HER1, HER2, and HER4)	[82]
	Neratinib	Inhibitor of the activity of the human EGFR family	Irreversibly binds to Cys-773 and Cys-805 of the ATP-binding domain of EGFR proteins (EGFR/HER1, HER2, and HER4), as well as downstream pathways including ERK and Akt.	[83]
	Osimertinib	Inhibitor of specific mutated forms of EGFR, including T790M, L858R, and exon 19 deletion	Irreversibly binds to Cys-797 of certain mutant forms of EGFR (L858R, exon 19 deletion, and double mutants containing T790M	[84]
EGFR Inhibitors		HER-2, HER-3, and HER-4. EGFR, ErbB-4,	Downregulates ErbB signaling	[85]
			PI3K/Akt/	
			Directed against the ErbB family of receptors	[86]
PI3K/AKT/mTOR pathway inhibitors	Buparlisib	pan-class I phosphoinositide 3-kinase isoforms inhibitor	Specifically inhibits class I PI3K and tubulin	[87]
	Alectinib	Tyrosine kinase inhibitor of ALK and RET proteins	electively inhibits the activity of ALK tyrosine kinase and tyrosine-protein kinase receptor RET.	[88]
	Crizotinib	Multikinase inhibitor	Competitively binds to the ATP-binding pocket of many receptor tyrosine kinases including ALK, Hepatocyte Growth Factor Receptor (HGFR, c-Met), reactive oxygen species 1 (ROS1), and Recepteur d’Origine Nantais (RON)	[89]
	Ceritinib	Potent inhibitor of ALK	Blocks autophosphorylation of ALK, which inhibits downstream signaling proteins.	[90]
Ras/Raf/MEK/ERK signaling pathway Inhibitors	Sorafenib	Multikinase inhibitor	Protein kinase inhibitor of many proteins, including VEGFR, PDGFR, and RAF kinases.	[91]
	Vemurafenib	Inhibition of the mutated BRAF V600E kinase	Inhibits a serine-threonine protein kinase B-RAF	[92]
	Dabrafenib (TAFINLAR^®^)	Inhibitor of B-raf protein	Inhibitor of the B-RAF and C-RAF proteins through ATP competitive binding of the active conformation of BRAF kinase	[93]
	Trametinib (Mekinist^®^)	Reversible allosteric inhibitor of MEK1 and MEK2 activity	It is an ATP non-competitive inhibitor that binds MEK adjacent to the ATP binding site in common with other MEK allosteric inhibitor	[94]
Immunomodulators	Nivolumab (Opdivo^®^)	MoAbs that bind to the PD-1 receptor	Bind to the PD-1 receptor and block its interaction with PD-L1 and PD-L2	[95,96,97,98,99]
	Pembrolizumab (Keytruda^®^)			[95,96,98,100]
	Durvalumab (Imfinzi^®^)	MoAbs that block the interaction of PD-L1	Block the interaction of PD-L1 with PD-1 and CD80 (B7. 1) to release the inhibition of immune responses.	[95,101,102,103]
	Atezolizumab (Tecentriq^®^)			[104,105]
	Avelumab (Bavencio^®^)			[106]
	Ipilimumab (Yervoy^®^)	MoAbs that bind to the CTLA-4 receptor	Binds to CTLA-4, blocking the inhibitory signals of T-cell inactivation.	[95,96,97,101,106,107]
	Tremelimumab			[95,101,102]

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
