# Peer review of "Overview of New Treatments with Immunotherapy for Breast Cancer and a Proposal of a Combination Therapy"

_molecules, 2020, doi:10.3390/molecules25235686_

Round 1

Reviewer 1 Report

The manuscript "Overview of new treatments with immunotherapy for
3 breast cancer and a proposal of combination therapy" by Morales at al. aims to summarize the current immunotherapeutic approaches and to propose the novel solutions that can be utilized in the treatment of this type of modality. 

The summarizing part is fine but I was a bit disappointed with the last paragraph as it points relatively little little.

Also, there are some sentences that are difficult to understand because of the grammar issues such as:

Lines 20-22:

However, current evidence shows that the use of
these new therapies is not always efficient, certainly because, in addition to the need to reactivate the Th1 immune response, it is also necessary the changing tumor microenvironment and coactivation of more components of the immune response.

or

469-471

Because more than 15% of TNBC have been associated with germline mutations in the BRCA1/2 (gBRCA), it has been supposed that could be sensitive to DNA-damaging agents such as cisplatin, carboplatin, and Poly(ADP-ribose) polymerase inhibitors (PARPi).

Please, check the manuscript carefully.

Line 67-69

What are those significant results?? Please, bo more precise.

Figure 1 - correct trastuzumab

Discuss if chapters 8 and 9 could be combined.

Author Response

Point 1: The summarizing part is fine but I was a bit disappointed with the last paragraph as it points relatively little little.

Response1: Line 24. The last paragraphs of the abstract were modified.

Point 2: Also, there are some sentences that are difficult to understand because of the grammar issues such as,

Before: Lines 20-22, However, current evidence shows that the use of these new therapies is not always efficient, certainly because, in addition to the need to reactivate the Th1 immune response, it is also necessary the changing tumor microenvironment and coactivation of more components of the immune response.

Response 2:

After: Line 24-32, Based on this knowledge, we formulate a proposal for the implementation of combined therapy using an extracorporeal immune response reactivation model and cytokines plus modulating antibodies, for co-activation of the Th1-and NK-dependent immune response, either in situ or through autologous cell-therapy.

Point 3: Before line 66-69, after line 81 (What are those significan results?)

Response 3: Added, in patients receiving neoadjuvant chemotherapy (NAC) vs adjuvant chemotherapy (AC), and chemotherapy was eliminated.

Appointment 8 was replaced:

Meng, L.; Ma, P. Apparent diffusion coefficient value measurements with diffusion magnetic resonance imaging correlated with the expression levels of estrogen and progesterone receptor in breast cancer: A meta-analysis. J Cancer Res Ther. 2016, 1, 36-42.

By:

Melchior, NM.; Sachs, DB.; Gauvin, G.; Chang, C.; Wang, ChD.; Sigurdson, ER.; et al. Treatment times in breast cancer patients receiving neoadjuvant vs adjuvant chemotherapy: Is efficiency a benefit of preoperative chemotherapy? Cancer Med. 2020, 8, 2742-51.

Point 4: The lines where they where modified have been modified by what was previously in line 469-471. Now it's in line 502-505.

Because more than 15% of TNBC have been associated with germline mutations in the BRCA1/2 (gBRCA), it has been supposed that could be sensitive to DNA-damaging agents such as cisplatin, carboplatin, and Poly(ADP-ribose) polymerase inhibitors (PARPi).

Respuesta 4:

One of them is associated with germline mutations in BRCA1/2 (gBRCA), this alteration occupies more than 15% of TNBC, as well as it has been proposed that they could be sensitive to DNA-damaging agents such as cisplatin, carboplatin and poly (ADP-ribose) polymerase inhibitors. (PARPi).

Point 5:

Figure 1 - correct trastuzumab

Response 5: Modified from traztusuma to traztusumab (line 207).

Point 6: Discuss if chapters 8 and 9 could be combined.

Respuesta: Línea 567: 8. A proposal of combination immunotherapy

Reviewer 2 Report

All concerns have been addressed. Notably, the manuscript contains excellent figures. However, there are still some typos including: line 22 “CD8+ T-cell”, line 27 “Th1- and NK-dependent immune response”, line 49 “(TNBC)”, line 351 “nivolumab (anti-PD-1 mAb) plus ipilimumab (anti-CTLA-4 mAb) OR “nivolumab (targets PD-1) plus ipilimumab (targets CTLA-4)”, etc.

Author Response

Response to Reviewer 2 Comments

Point 1: Line 22, changed to:

Response 1: Before, T CD8 cell. After, CD8+ T cell

Point 2: Line 27, changed from:

Response 2: Th1 and NK immune response

To:

Th1- and NK-dependent immune response

Point 3) Line 33, we include in abbreviations TNBC Triple-Negative Breast Cancer.

Response 4:

Line 48-49: Changed from triple-negative BC to Triple-Negative Breast Cancer and removed parentheses and bolds.

Before; the triple-negative BC (TNBC) (ER-, PR- and HER-2-).

After: the Triple-Negative Breast Cancer (TNBC: ER-, PR- and HER-2-).

Point 5: There are still some typos including

Response 5: Line 355-357, changed from: nivolumab (anti-PD-1 mA) plus ipilimumab (anti-CTLA-4-mAb) in TNBC (NCT01928394), and in a single-arm phase II trial of durvalumab plus tremelimumab (anti-CTLA-4-mAb).

This manuscript is a resubmission of an earlier submission. The following is a list of the peer review reports and author responses from that submission.

Round 1

Reviewer 1 Report

The paper entitled "From current experience in breast cancer treatments toward a proposal of combination therapy" is a comprehensive and well-organized review that focus on modalities used or tested in breast cancer settings.

I do not have any remarks on the content as I read the paper with interest.

Some of the Figures could be enhanced to meet the quality of the text. Figure 2 is a bit difficult to read and out of style when compared to the other Figures.

Figure 4 is also unclear and for sure not readable without text.

Author Response

Q1. Concerning Figure 2, this was difficult to understand:
Reply. Thank you very much for your comment. We put the therapeutic targets and made annotations so that the figure is understood without reading the text?
Q2. Figure 4a and 4b were also modified to be understood without reading the text, and they were renumbered to 4 and 5.

Reviewer 2 Report

Concerns and recommendations for authors:

1) The title of the manuscript "From current experience in breast cancer treatments  toward a proposal of combination therapy" implies discussion of all types of BC and all treatment strategies. In this concern it is recommended to (i) give classification of BC and (ii) change in the title the word "treatment" for "chemotherapy and immunotherapy" since the authors do not discuss other treatment strategies such as radiotherapy and surgery

2) In relation to 1) it is recommended to discuss treatment not only of HER2-positive BC, but also ER-positive and PR-positive BC.

3) In this concern, it is also recommended to re-write Abstract and Introduction.

4) It is also recommended to describe more experimental details and clinical trial data. References contain only few latest articles published in 2019-2020. More recent data should be discussed.

5) The manuscript is not well-organized. For example, in section "Therapeutic antibodies in breast cancer" the authors discuss predominantly anti-HER2 Mabs. Therefore, section 3 should be renamed as "HER2 targeting therapeutic antibodies in breast cancer".

6) Since PARP is not a component of signal transduction pathways, PARPI should be moved from section 5 to another section.

7) In section 5 "Signal transduction inhibitors" the authors discuss inhibitors of signaling pathways. However, inhibitors of some pathways such as PI3K/AKT/mTOR pathway have not been discussed.

8) Also, it is recommended to combine sections 7, 8 and 9.

9) Section 2 "The immune system and cancer" is very short. Additional discussion should be added.

10) Fig. 1: arrows have different width, correct “tamoxifen”.

11) Fig. 2 looks fine. However, it is recommended to add molecular targets for olaparib and doxorubicin to make the figure more informative.

12) Table 1 is recommended to be divided into three Tables and to place them in different sections according to type of a drug:

13) It is recommended to unify the abbreviations, i.e. MoAbs or Mabs),

14) It is recommended to add to Table 1 a column for molecular effects of a drug.

15) Edit English language and check grammar: for example on lines 38, 46, 163, 174, 176, 184, 185, 189, 194, 429 macrophages ((MΦs, etc.

Author Response

1) The title of the manuscript "From current experience in breast cancer treatments toward a proposal of combination therapy" implies discussion of all types of BC and all treatment strategies.
Reply. Thank you very much for your comment. We have also changed the title to: "Summary of new immunotherapy treatments for breast cancer and a proposal for combination therapy". (i) BC Classification included in Introduction. (ii) It was done by changing the title.
2) In relation to 1) it is recommended to discuss treatment not only of HER2-positive BC, but also ER-positive and PR-positive BC.
Reply. Thank you very much for your comment The manuscript presents more immunotherapy for triple negative breast cancer, but we have included it for HER-2, ER and PR positive.
3) In this concern, it is also recommended to re-write Abstract and Introduction.
Reply. Thank you very much for your comment. We have modified the introduction and the background part to make it more concise and relevant to the content of our study by adding more information about ER and PR positive tumors. Current treatment for ER and PR tumors was also included (line 45-79).

4) It is also recommended to describe more experimental details and clinical trial data. References contain only few latest articles published in 2019-2020. More recent data should be discussed.
Reply. Indeed, most appointments are between 2019 and 2020. We consider that the manuscript is very extensive; therefore, the reader was able to consult all the work in the reference.
5) The manuscript is not well-organized. For example, in section "Therapeutic antibodies in breast cancer" the authors discuss predominantly anti-HER2 Mabs. Therefore, section 3 should be renamed as "HER2 targeting therapeutic antibodies in breast cancer".
Reply Thank you very much for your comment. The manuscript was reorganized, many segments were re-accommodated. Section 3 is changed to HER2 targeting therapeutic antibodies in breast cancer (page 4, line 140).
6) Since PARP is not a component of signal transduction pathways, PARPI should be moved from section 5 to another section.
Reply. Thank you very much for your suggestions. The PARP was moved to the section 7.-Other inhibitors used in chemotherapy
7) In section 5 "Signal transduction inhibitors" the authors discuss inhibitors of signaling pathways. However, inhibitors of some pathways such as PI3K/AKT/mTOR pathway have not been discussed.
Reply. Thank you very much for your suggestions. It is already included IP3K/AKT/mTOR in section 4. Signal transduction inhibitors.
8) Also, it is recommended to combine sections 7, 8 and 9.
Reply. Sections 7 and 8 were combined.
9) Section 2 "The immune system and cancer" is very short. Additional discussion should be added.
Reply. Section 2. Immune system and cancer, were enlarged including antigen recognition and activation of cancer response. No editing is included, only mentioned.
10) Fig. 1: arrows have different width, correct “tamoxifen”.
Reply. Tamoxifeno was corrected to Tamoxifen and the size of the arrow.
11) Fig. 2 looks fine. However, it is recommended to add molecular targets for olaparib and doxorubicin to make the figure more informative.
Reply. Targets were set in Figure 2 for Olaparib, doxorubicin, and cetuximab.
12) Table 1 is recommended to be divided into three Tables and to place them in different sections according to type of a drug:
Reply. The original table 1 was divided into 3 new tables and placed near the corresponding text or sections.
13) It is recommended to unify the abbreviations, i.e. MoAbs or Mabs),
Reply. The abbreviation looked like; MoAb to mention monoclonal antibodies.
14) It is recommended to add to Table 1 a column for molecular effects of a drug.
Reply. Thank you very much for your comment. The table already contains white and molecular mechanism of action.
15) Edit English language and check grammar: for example on lines 38, 46, 163, 174, 176, 184, 185, 189, 194, 429 macrophages ((MΦs, etc.
Reply. Thank you very much for your suggestions. English language and grammar were revised for suggested paragraphs.

Reviewer 3 Report

The present review aimed at summarizing the pros and cons of the main immunotherapeutic strategies for the treatment of breast cancer (therapeutic mAbs, signal transduction inhibitors, cytokines, etc..), focusing also on the new approaches concerning the immune checkpoint blockade (anti-PD-1 and anti-CTL4 mAbs). The ultimate goal is to suggest some combination of different immunotherapeutic strategies to overcome the heterogeneity observed in breast cancer subtypes.

Although the aim of the review is sometimes not clear-cut (i.e. Abstract section), it is very ambitious, but the overall result is superficial, especially as a consequence of poor in-depth analysis of some themes, or lack of some updates about recent discoveries or innovative therapies. Just to give some examples, the authors pretended to focus on breast cancer, but without discerning in detail breast cancer subtypes, whose molecular and immunological features and peculiarities are mandatory for understanding the success and fails of most immunotherapeutic strategies in clinical use or development. The authors pointed out the importance of NK cells as weapons to implement breast cancer immunotherapy, but they do not mentioned the possibility of validating Chimeric Antigen Receptor-NK Cells, neither described the recent identification of the PD-1+ NK subset that opened up the possibility of exploring the PD-1/PD-L1 axis and immune-checkpoint inhibitors in this ground-breaking context.

Overall, this review does not provide an advance towards the current knowledge in breast cancer immunotherapy, which have been already analysed in more detail in other high quality and elegant peer-reviewed works (just to cite some examples: García-Aranda M and Redondo M. Cancers (Basel). 2019 Nov 20;11(12):1822. doi: 10.3390/cancers11121822; Force J et al. Curr Treat Options Oncol. 2019 Mar 28;20(4):35. doi: 10.1007/s11864-019-0634-5; Solinas C et al. ESMO Open. 2017 Nov 14;2(5):e000255. doi: 10.1136/esmoopen-2017-000255. eCollection 2017;  Chrétien S et al. Cancers (Basel). 2019 May 5;11(5):628. doi: 10.3390/cancers11050628).

English language is not of high level, sometimes elementary in sentence construction. The lexical quality is limited, with frequent repetitions of the same terms in contiguous sentences (i.e., “cases” repeated three times, lines 39 and 40; “mortality” repeated three times; lines 42 and 43; “therapies” reported twice in the same line #59, etc…). I therefore suggest revising the text enriching it with synonyms, or rewriting and connect sentences in order to avoid repetitions. There are also grammar errors, as for example in the sentence “These treatments could be targeted locally or regional and systemic.” (lines 53-54). The review also includes misprints (i.e., in Table 1 references are presented in more than one style). Overall, I suggest an extensive revision of the entire text, with particular care of editing English language and style.

Additionally, abstract should be revised since some concepts reported are too much vague, resulting puerile. Specifically, the sentence “Therefore, combining therapies with the coordinated activation of each cell of the immune response will probably produce better results.” (lines 21-22), seems to not take into account the complexity of the anti-tumor immune responses, without either defining which “cells” and cell subclasses of the immune system are taken into consideration. Another sentence is “In this article, recent advances in the treatment of cancer aimed at blocking signaling pathways and the use of monoclonal antibodies directed to receptors were reviewed” (lines 25-26) is again too much vague, and does not reflect the real aim of the work, which tempted to focus on breast cancer. The aim of the review in the Abstract section remains nebulous. Also the introduction is too much vague, and does not appropriately focus on breast cancer and breast cancer subtypes, which in my opinion is mandatory for the introduction of all the immunotherapeutic strategies then listed. 

Author Response

English language and grammar were revised for full-manuscript editing as required.

Summary and introduction were modified to exemplify the subtypes of breast cancer, and the purpose of the manuscript was to be more towards triple-negative therapy.

Q1. Although the aim of the review is sometimes not clear-cut (i.e. Abstract section), it is very ambitious, but the overall result is superficial, especially as a consequence of poor in-depth analysis of some themes, or lack of some updates about recent discoveries or innovative therapies. Just to give some examples, the authors pretended to focus on breast cancer, but without discerning in detail breast cancer subtypes, whose molecular and immunological features and peculiarities are mandatory for understanding the success and fails of most immunotherapeutic strategies in clinical use or development.
Reply: Breast cancer subtype classification, has been extended, making emphasis on its molecular characteristics.
Q2. The authors pointed out the importance of NK cells as weapons to implement breast cancer immunotherapy, but they do not mentioned the possibility of validating Chimeric Antigen Receptor-NK Cells, neither described the recent identification of the PD-1* NK subset that opened up the possibility of exploring the PD-1/PD-L1 axis and immune-checkpoint inhibitors in this ground-breaking context.
Reply: An update has been made on the role of NK PD-1/PD-L1 cells and their therapeutic potential in breast cancer therapy.
Q3. Overall, this review does not provide an advance towards the current knowledge in breast cancer immunotherapy, which have been already analysed in more detail in other high quality and elegant peer-reviewed works.
Reply: We understand the reviewer's suggestion about other review articles that address this topic. However, we consider the large amount of information written in our paper is enough to illustrate the need to use personalized medicine with combination therapies to improve the survival rate in cancer patients.
Q4. English language is not of high level, sometimes elementary in sentence construction. The lexical quality is limited, with frequent repetitions of the same terms in contiguous sentences. …. Overall, I suggest an extensive revision of the entire text, with particular care of editing English language and style.
Reply:  English editing and correction of grammatical mistakes were done.
Q5. Additionally, abstract should be revised since some concepts reported are too much vague, resulting puerile. Specifically, the sentence "Therefore, combining therapies with the coordinated activation of each cell of the immune response will probably produce better results." (lines 21-22), seems to not take into account the complexity of the anti-tumor immune responses, without either defining which "cells" and cell subclasses of the immune system are taken into consideration. Another sentence is 'in this article, recent advances in the treatment of cancer aimed at blocking signaling pathways and the use of monoclonal antibodies directed to receptors were reviewed" (lines 25-26) is again too much vague, and does not reflect the real aim of the work, which tempted to focus on breast cancer. The aim of the review in the Abstract section remains nebulous. Also the introduction is too much vague, and does not appropriately focus on breast cancer and breast cancer subtypes, which in my opinion is mandatory for the introduction of all the was corrected to immunotherapeutic strategies then listed.
Reply: As suggested by the reviewer, both the abstract and the introduction were modified and updated.

“We would like to thank the referees again for taking the time to review our manuscript.”

Best Regards

Teran and Cols.